# Effect of Three-Dimensional-Printed Thermoplastics Used in Sensor Housings on Common Atmospheric Trace Gasses

**DOI:** 10.3390/s24082610

**Published:** 2024-04-19

**Authors:** Tristalee Mangin, Evan K. Blanchard, Kerry E. Kelly

**Affiliations:** Department of Chemical Engineering, University of Utah, Salt Lake City, UT 84112, USA

**Keywords:** air quality, low-cost sensors, sensor housing, 3D printing, thermoplastic polymers

## Abstract

Low-cost air quality sensors (LCSs) are becoming more ubiquitous as individuals and communities seek to reduce their exposure to poor air quality. Compact, efficient, and aesthetically designed sensor housings that do not interfere with the target air quality measurements are a necessary component of a low-cost sensing system. The selection of appropriate housing material can be an important factor in air quality applications employing LCSs. Three-dimensional printing, specifically fused deposition modeling (FDM), is a standard for prototyping and small-scale custom plastics production because of its low cost and ability for rapid iteration. However, little information exists about whether FDM-printed thermoplastics affect measurements of trace atmospheric gasses. This study investigates how five different FDM-printed thermoplastics (ABS, PETG, PLA, PC, and PVDF) affect the concentration of five common atmospheric trace gasses (CO, CO_2_, NO, NO_2_, and VOCs). The laboratory results show that the thermoplastics, except for PVDF, exhibit VOC off-gassing. The results also indicate no to limited interaction between all of the thermoplastics and CO and CO_2_ and a small interaction between all of the thermoplastics and NO and NO_2_.

## 1. Introduction

Low-cost gas sensors are becoming a popular option for air quality measurement applications [1,2,3,4]. These sensors offer a cost-effective, compact, portable alternative to traditional monitors. Additionally, low-cost sensors have enabled the creation of large sensor networks that augment established government monitoring sites and increase the spatial distribution of measurements [5,6,7,8]. Simultaneously, 3D printing, particularly fused deposition modeling (FDM) using thermoplastic filaments, has emerged as an economical means to craft customized structural supports and sensor housings for air quality research applications [9,10,11,12,13,14,15,16,17].

FDM, an additive manufacturing technique, involves sequentially adding layers of material to produce a three-dimensional object [9,18]. An FDM-printed component begins as a 3D model containing part geometry and dimensions, which is processed by slicing software and sent to the 3D printer, where the filament is fed through a nozzle heated above the glass transition temperature of the polymer, referred to as the extrusion temperature throughout this study, and extruded to produce the 3D-printed component [9,18]. However, to the best of our knowledge, few studies have explored the effects of FDM thermoplastics on trace gas concentrations in air quality sensor housing applications.

Thermoplastics can off-gas volatile organic compounds (VOCs) through several mechanisms: vaporization of the thermoplastic or additives, desorption of gasses or manufacturing solvents from the surface, diffusion of dissolved gasses or solvents inside the solid, and thermal or chemical degradation over time [19,20,21]. Multiple studies have evaluated vacuum the off-gassing of VOCs from commercially manufactured thermoplastic products using pressure rise measurements [20,21] or total mass lost measurements [22] and analyzed the chemical composition using various forms of mass spectroscopy [21,22,23]. NASA has compiled a database of material vacuum off-gassing using a standard test method that reports total mass lost and collected volatile condensable materials [24]. However, these studies are difficult to translate into impacts on low-cost air quality sensor systems, which typically operate at atmospheric pressure. Pressure and mass measurements from vacuum-based studies provide the total off-gassing of all chemical species, not just VOCs. Mass spectra intensities can be used to compare relative VOC off-gassing levels across thermoplastics. However, concentration calibration curves would need to be provided with the mass spectra results to convert the intensities to concentrations [25]. Using mass spectroscopy, a study by Zwicker et al. [26] analyzed VOC off-gassing in a vacuum from FDM-printed polylactic acid (PLA) and reported an increase in VOC off-gassing at temperatures greater than 75 °C, as indicated by increased mass spectra signal intensities. Budde et al. [27] investigated off-gassing at atmospheric pressure using ion mobility spectrometry/mass spectrometry for commercial thermoplastics, including polycarbonate (PC), polyvinylidene fluoride (PVDF), and acrylonitrile butadiene styrene (ABS), and found that ABS had the highest VOC off-gassing and fluoropolymers (including PVDF) had the lowest VOC off-gassing levels, as indicated by comparing the mass spectra signal intensities.

Table 1 summarizes several studies that have examined VOC emissions at atmospheric pressure from FDM-printed thermoplastics during the FDM printing process using various methods [28,29,30,31,32,33,34,35]. Some of these studies provided somewhat contradictory rankings of VOC emissions from different thermoplastic filaments, likely due to different VOC measurement methods and FDM settings. Additionally, only two of these studies continued to measure VOC emissions for 1–2 h after completion of the FDM print [32,33], and air quality applications that would use FDM-printed thermoplastic components occur well after print completion. None of the studies analyzed VOC emissions from PC or PVDF.

To the best of our knowledge, no studies have examined carbon monoxide (CO), carbon dioxide (CO_2_), nitrogen monoxide (NO), and nitrogen dioxide (NO_2_) gas reactivity with ABS, PLA, PETG, PC, and PVDF. However, multiple studies have examined the reaction between nitrogen oxides (NO_x_) and various thermoplastics, and these studies can shed light on NO_x_ reactions with different groups of thermoplastics [36,37,38]. NO_2_ is a strong oxidizing agent [38,39]. Polymer structures that are particularly sensitive to reactions with NO_2_ include those that contain carbon–carbon double bonds, amide groups (polyamides, polyurethanes, and polyamidoimides), and peroxyl macroradicals [37]. Thermoplastics with no carbon–carbon double bonds and no functional groups sensitive to NO_2_ exhibit little reaction with NO_2_ [37]. NO can act as either an oxidizing or a reducing agent, and NO reacts with O_2_ to form NO_2_ [37,40]. NO is not able to remove tertiary or allylic hydrogens from organic molecules and is not able to add to isolated double bonds, but NO readily reacts with free radicals [37]. Additionally, Parriskii et al. hypothesized that NO-diene reactions stem from NO_2_ mixed with the NO, particularly if the NO is in the presence of O_2_ [37]. Finally, these reaction studies focused on polymer degradation versus the impact on gas concentration.

This study evaluated the interaction between FDM-printed thermoplastic baffles—ABS, PLA, PETG, PC, and PVDF—and key atmospheric trace gasses—CO, CO_2_, NO, NO_2_, and VOCs—at atmospheric pressure using research-grade air quality instruments. In the subsequent sections of this paper, the FDM-printed thermoplastic baffles will be referred to as “baffles”. ABS, PLA, and PETG were chosen for this study as they are some of the most common FDM thermoplastic filaments [18]. PC and PVDF were added to this study because these are commercially available FDM thermoplastic filaments and are expected to have small reactivity with NO_2_ based on their chemical structures [37,38,41,42,43]. Additionally, PVDF is a fluoropolymer, which was reported to have low VOC off-gassing compared with other thermoplastics [27]. Appendix A shows the chemical structures for each of the five thermoplastics. CO, CO_2_, NO, NO_2_, and VOC were chosen for this study because the ultimate application is to measure air quality metrics in an area with combustion emissions. Each of the five baffles was initially exposed to zero air to determine whether the thermoplastic off-gassed any of the target trace gasses. If no off-gassing was detected, then the baffle was exposed to varying concentrations of the target gas to identify possible reactions between the thermoplastic and the gas.

The results of this study indicated that after completion of the FDM-printing process, ABS, PLA, PETG, and PC still release VOCs, while PVDF releases little to no VOCs. Additionally, the results indicated that CO and CO_2_ have little to no reaction with any of the thermoplastics, but NO and NO_2_ both showed reactivity toward the five thermoplastics. These results provide important considerations for researchers designing air quality measurement setups employing FDM-printed components.

## 2. Materials and Methods

This study analyzed the effects of five different thermoplastics on five common atmospheric trace gasses. The experimental setup and design remained the same for the off-gassing and reaction experiments, but the gas concentrations and the analyses differed. Appendix A contain an experimental matrix for each of the gasses. Appendix A contains a list of all equipment used in the study. Ultimately, our team aims to use FDM-printed thermoplastic structural components inside an enclosure equipped with a small fan to continuously draw air through the enclosure in an air quality application measuring CO_2_, CO, NO, NO_2_, and VOCs. Consequently, the off-gassing and reaction experiments were performed using a constant flow system.

### 2.1. Experimental Setup

Figure 1 shows the experimental setup. Table 2 lists the research-grade instruments for measuring gas concentrations and relevant specifications for each instrument. The experimental setup used was a 1500 cm^3^ (15 cm × 10 cm × 10 cm) acrylic chamber that acted as the reaction volume. The chamber was assembled using acrylic cement. The chamber had a detachable lid that was closed using a Teflon liner and c-clamps. The chamber operated at atmospheric pressure with a constant 1100 (±1.12) ccm total volumetric flow rate, which provided sufficient flow for each research-grade instrument (Table 2) and resulted in a gas residence time of approximately 75 s. The baffles for each thermoplastic were sequentially placed inside the chamber where the reaction between the gas and the thermoplastic could occur. A dilution system mixed gas from a calibration cylinder with air from either a zero-air cylinder or zero-air generator (Teledyne High-Performance Zero Air Generator-Model T701H) that provided a constant flow of the target gas concentration. The gas flow was controlled with mass flow controllers (MFCs), and all equipment was connected using a 1/4-inch diameter polytetrafluoroethylene (PTFE) tubing and stainless steel fittings. Flows in the chamber and into the research-grade instruments were laminar (Re < 250). Chamber temperature and relative humidity were monitored with a DHT22 sensor located in the chamber.

Figure 2 provides a picture of the baffles. Table 3 lists the baffle and FDM printer characteristics. Appendix A provide a list of the manufacturers for each FDM printer filament and the details of the baffle design and FDM printer settings, respectively. Appendix A show images of the base plate and vertical baffle from Fusion360. Table 3 lists the date each thermoplastic baffle was FDM-printed. Appendix A list the off-gassing and reaction experiment dates. Experiments were not started until several days after the FDM print so that the baffles were in thermal equilibrium with ambient laboratory temperature for all experiments.

### 2.2. Experimental Design

Table 4 summarizes the gas concentrations for the off-gassing and reaction experiments. All off-gassing experiments occurred in zero air. For reaction experiments, two initial concentrations were completed for each gas. If a reaction occurred, then a third concentration was performed to estimate reaction kinetics. Regardless of whether the experiment was for off-gassing or reaction, each gas concentration consisted of three phases (Figure 3). In the first phase, the chamber remained empty with no baffle inside, and baseline concentration measurements were recorded for ten minutes after the research-grade instrument indicated a steady-state gas concentration. In the second phase, the reaction chamber was opened, and the baffles were placed inside. The chamber was closed. The gas concentration was allowed to reach a steady state, and concentration measurements were recorded for ten minutes. In the third phase, the chamber was opened, and the baffles were removed. The chamber was closed. The gas concentration was allowed to reach a steady state, and a second 10 min period of baseline concentration measurements were recorded with no baffles inside the chamber. This approach ensured that the baseline concentration measurements in phases one and three considered any interactions with the acrylic chamber.

### 2.3. Off-Gassing Data Analysis

Off-gassing measurements from the experimental matrices (Appendix A) were evaluated to determine whether a baffle was off-gassing the target gas. It was assumed that the gas density was constant, the gas did not accumulate in the chamber, and mass transfer effects within the solid thermoplastic were negligible. Using these assumptions, a material balance for the reactant gas with a constant total flow rate produced:(1)$$-r_{A}=\frac{F(C_{A0}-C_{A1})}{SA}.$$
where $-r_{A}$ is the reaction rate of the gas; *F* is the gas flow rate; $C_{A0}$ is the measured baseline gas concentration without the baffles in the chamber; $C_{A1}$ is the measured gas concentration with the baffles in the chamber; and SA is the surface area of the baffle. Notice that for a gas that is reacting with the baffle, $-r_{A}$ (Equation (Equation 1)) will be positive (positive reaction rate). Meanwhile, for a baffle that is off-gassing, $-r_{A}$ (Equation (Equation 1)) will be negative (negative reaction rate). Previous studies using the pressure rise method to measure vacuum off-gassing of thermoplastics report the off-gassing rates per unit surface area [21,23].

The error in the reaction rate (Equation (Equation 1)), or off-gassing rate, was estimated using the measurement error and equipment uncertainty:(2)$$e_{-r_{A}}=\sqrt{{\left(\frac{\left(C_{A0}-C_{A1}\right)}{SA}e_{F}\right)}^{2}+{\left(\frac{F}{SA}e_{\left(C_{A0}-C_{A1}\right)}\right)}^{2}+{\left(-\frac{F*(C_{A0}-C_{A1})}{SA^{2}}e_{SA}\right)}^{2}}.$$
where $e_{-r_{A}}$ is the estimated error of the reaction rate; $C_{A0}-C_{A1}$ is the difference in the measured concentrations with and without the baffles; $e_{\left(C_{A0}-C_{A1}\right)}$ is the error in the difference of the measured concentration with and without the baffles; $e_{F}$ is the uncertainty in the flow rate provided by the MFCs; and $e_{SA}$ is the approximate error in the surface area of the baffles. The error in the difference of the measured concentrations, $e_{\left(C_{A0}-C_{A1}\right)}$, was determined by adding the standard deviation of the two sets of measured concentrations together. Appendix A includes the calculation for the uncertainty in the MFC flow rate, and Appendix A includes the calculation for the error in the surface area of the baffles.

All data analysis was performed in Python 3 using the Pandas, Numpy, and SciPy libraries [55,56,57,58]. An analysis of variance (ANOVA) test was performed to determine if the difference in the two concentration measurements ($C_{A0}$ and $C_{A1}$) was statistically significant. The ANOVA analysis used “scipy.stats.f_oneway”. For the ANOVA analysis, a *p*-value less than 0.05 was considered statistically significant. Additionally, null hypothesis significance testing (NHST) was conducted between each thermoplastic’s VOC off-gassing results. NHST used a Bonferroni adjusted significance level of 0.0025 based on running 20 NHSTs. Appendix A discusses the calculation for NHST.

### 2.4. Reaction Data Analysis

In reaction experiments, the measured concentration difference between $C_{A0}$ and $C_{A}$ from the experimental matrices (Appendix A) was used to estimate the reaction rate (Equation (Equation 1)) per unit surface area at each target gas concentration. Similar to off-gassing, the ANOVA test was also performed to determine if the difference in concentration measurements ($C_{A0}$ and $C_{A1}$) was statistically significant. If the two initial target gas concentration experiments resulted in statistically significant concentration differences and in positive reaction rates with estimated errors that did not span zero, then a third concentration was completed for the gas and baffle. Reaction kinetics were analyzed.

The reaction rate kinetic analysis relied on the following key assumptions. First, it assumed that the thermoplastic was not consumed in the reaction. Second, as FDM-printing approaches tend to produce 3D-printed objects with low porosity [18], it assumed that the reaction only occurs on the surface of the baffle, and there are sufficient surface reaction sites such that mass transfer effects are neglected. Appendix A discusses the baffle surface area and surface area uncertainty calculations. Third, it assumed that the reaction is irreversible. These assumptions led to the following reaction equation
(3)$$aA_{gas}\to cC_{gas,gas-solid}.$$
where *A* is the reactant gas; *C* is the gas or gas–solid product; *a* is the reactant gas stoichiometric coefficient; and *c* is the product stoichiometric coefficient. With the assumption that the thermoplastic surface area (SA) was in excess, the kinetic equation reduced to
(4)$$-r_{A}=kC_{A}^{\alpha }.$$
where $-r_{A}$ is the reaction rate of the reactant gas; $C_{A}$ is the reactant gas concentration; *k* is the reaction rate constant; and α is the reaction order.

Nonlinear regression of Equation (Equation 4) provided estimates of the reaction rate constant and reaction order. The nonlinear fit was performed using “scipy.optimize.curve_fit”, keeping the method argument default such that the Levenber–Marquardt algorithm was applied in the least-squares analysis. Additionally, a 95% confidence interval was computed for both regression parameters. The covariance matrix from “scipy.optimize.curve_fit” was used to compute the 95% confidence interval for each of the parameters,
(5)$$CI=X\pm Z*X_{SE}.$$
where *Z* is the standard normal distribution quantile of (1−CI)/2 with a CI of 0.95 computed using “scipy.stats.norm.ppf”; *X* is the optimal parameter array; and $X_{SE}$ is the parameter standard error computed from the covariance matrix
(6)$$X_{SE}=\sqrt{VAR(X)}.$$

## 3. Results and Discussion

This study examined the interaction between five FDM-printed thermoplastics—PLA, ABS, PETG, PC, and PVDF—and five common atmospheric trace gasses—CO, CO_2_, NO, NO_2_, and VOCs. The experiments were divided into two groups, off-gassing and reaction experiments. The off-gassing results indicated that all five of the thermoplastics exhibit VOC off-gassing. However, the reaction results revealed that CO and CO_2_ display negligible reaction rates with all five of the thermoplastics, while NO and NO_2_ exhibit small reaction rates with all five of the thermoplastics.

### 3.1. Off-Gassing Experiments

Appendix A present the results for the off-gassing experiments organized by gas. It should be noted that the research-grade instruments were operating near their detection limits during the CO, CO_2_, NO, and NO_2_ off-gassing experiments. The results suggest that all five thermoplastics exhibited negligible off-gassing of CO, CO_2_, NO, and NO_2_. Negligible off-gassing included: (1) measured concentrations of zero, (2) differences in measured baseline concentrations and concentrations with baffles that were not statistically significant, and (3) statistically significant concentration differences leading to low off-gassing rates, accompanied by large margins of error that encompassed zero. However, each of the thermoplastics exhibited VOC off-gassing, with PVDF having a VOC off-gassing rate two orders of magnitude lower than the rest of the thermoplastics.

Appendix A lists the VOC off-gassing results. The VOC off-gassing experiments exhibited concentrations above the TSI Q-Track’s detection limit (10 ppb) and showed statistically significant differences in baseline concentrations and concentrations with the baffles. Appendix A shows that the difference between the VOC baseline concentrations and VOC concentrations with PVDF were close to the significance value of 0.05 (*p* = 0.0393), while the ANOVA *p*-values for PLA, ABS, PETG, and PC were statistically significant (<0.0001). Figure 4 illustrates that PLA, ABS, PETG, and PC yielded VOC off-gassing rates ranging between 18 and 31 $\frac{\mathrm{ccm}*\mathrm{ppb}}{{\mathrm{cm}}^{2}}$. However, the VOC off-gassing rate for PVDF was approximately two orders of magnitude lower (0.4 $\frac{\mathrm{ccm}*\mathrm{ppb}}{{\mathrm{cm}}^{2}}$). Appendix A shows the NHST results between the VOC off-gassing results for each thermoplastic. These results suggest that all of the thermoplastics except for PVDF exhibit VOC off-gassing. Therefore, PVDF is a good candidate for FDM-printed thermoplastic components in applications focusing on low-concentration VOC measurements. However, PVDF necessitates an FDM printer capable of handling higher extrusion temperatures (Table 3) and FDM printing inside a gas hood [59]. These limitations may constrain the suitability of PVDF for certain applications.

Table 1 summarizes the results from previous studies that have examined VOC emissions during the FDM printing process. This study differs from these previous studies in its focus on VOC emissions after the completion of the FDM print. However, the relative magnitude of VOC emissions from the previous studies can still provide a source of comparison. Similar to the previous studies, this study also found that ABS had the highest VOC emission rate. However, this study found that PLA and PETG had similar VOC emission rates after FDM print completion, while the Wojnowski study found that PETG had a lower VOC emission rate than PLA during the FDM printing process. This study resulted in the same rankings for ABS, PC, and PVDF as the Budde et al. [27] study evaluating commercial products, where ABS had the highest VOC off-gassing levels and PVDF had the lowest VOC off-gassing levels.

### 3.2. Reaction Experiments

Appendix A present the reaction experiment results organized by gas. The five thermoplastics exhibited no to limited reaction rates with CO_2_ and CO, while NO and NO_2_ displayed measurable reaction rates. The VOC results showed measurable negative reaction rates, which indicate off-gassing. Appendix A includes graphs for each gas with the reaction rate results converted to molar units.

Appendix A and Figure 5a,b show CO and CO_2_ exhibiting low reaction rates with estimated errors that span zero, suggesting that none of the five thermoplastics react with CO and CO_2_. Consequently, we opted not to pursue a third concentration for CO and CO_2_. The results are in line with CO_2_ being a very stable molecule [60] and CO being a weak reducing agent [61]. PLA and ABS exhibited relatively easy printing characteristics with lower extrusion temperature requirements than PC and PVDF (Table 3) and are not as brittle as PETG [62]. Consequently, PLA and ABS are good candidates for air quality applications involving CO and/or CO_2_ measurements.

Appendix A list the results for NO and NO_2_ and show that both NO and NO_2_ exhibited statistically significant differences in baseline concentrations and concentrations with the baffles. Figure 6a,b indicate that both NO and NO_2_ exhibit small but measurable reaction rates with each of the thermoplastics. These reaction rates are concentration-dependent. Appendix A lists the parameters for the nonlinear fit of Equation (Equation 4). ABS had the largest reaction rates with both NO and NO_2_. As discussed in the Introduction, ABS contains functional groups (carbon–carbon double bonds, Appendix A) that are sensitive to reactions with NO_2_ [37]. PLA, PETG, PC, and PVDF do not contain functional groups that are considered sensitive to NO_2_ [37] (Appendix A) and are not expected to react with NO_2_ at the same level as ABS. PC and PVDF exhibited the lowest reaction rates with NO. However, when considering NO_2_, PLA and PETG appeared to have slightly smaller reaction rates than PC and PVDF. Therefore, any of the thermoplastics, except ABS, are possible candidates for FDM-printed components for air quality applications involving NO and/or NO_2_ measurements.

The reaction results for VOC (Appendix A and Figure 7) revealed statistically significant differences between baseline concentrations and concentrations with the baffles that resulted in negative reaction rates, which indicate VOC off-gassing (further discussed in the off-gassing results section).

### 3.3. Implications for Air Quality Measurements

To relate our experimental results to real-world air quality measurements, we calculated the difference in VOC, NO, and NO_2_ concentrations using this study’s off-gassing and reaction rates for two sets of theoretical scenarios. The first (called moderate and detailed in Appendix A) used an FDM area of 118 cm^2^, VOC concentration of 41 ppb, NO/NO_2_ concentrations of 35 ppb, and residence time of either 2 or 75 s. The second (called worst case and detailed in Appendix A) used an FDM surface area of 1478 cm^2^, VOC concentration of 10 ppb, NO concentration of 600 ppb, NO_2_ concentration of 200 ppb, and residence time of either 2 or 75 s. Although some concentrations used in these scenarios are lower than our experimental conditions, Davydov et al. [36] discussed that polymeric degradation at low gas concentrations could be extrapolated from results at higher gas concentrations. Consequently, we assumed constant VOC off-gassing rates and NO/NO_2_ reaction rate parameters (*k* and α).

For the moderate scenario, the thermoplastics caused VOC concentrations to increase by <1% (2 s residence time) to 4% (75 s residence time). However, for the worst-case scenario, VOC concentrations increased by <1% (PVDF) to 5% (ABS) at the 2 s residence time and 3% (PVDF) to 200% (ABS) at the 75 s residence time. FDM thermoplastics had a smaller effect on NO and NO_2_ concentrations. For the moderate scenario, the thermoplastics caused NO/NO_2_ concentrations to decrease by <1% (both the 2 and 75 s residence times). For the worst case scenario, NO and NO_2_ concentrations decreased by <1% for all thermoplastics at the 2 s residence time, and these concentrations decreased by 0.6% (NO with PC) to 3% (NO_2_ with ABS). These results indicate that VOC off-gassing from thermoplastics is a larger concern than NO/NO_2_ reactions with thermoplastics. VOC off-gassing could be a particular concern if the sensor relies on passive gas transport, has a long residence time, or seeks to measure low VOC concentrations (low ppb or ppt ranges).

### 3.4. Limitations and Future Work

This study has several limitations. First, the experimental design employed a constant gas flow setup with a residence time of approximately 75 s. Low-cost air quality sensing applications commonly employ a continuous flow setup, utilizing a fan to draw flow through a housing [12,63,64], but tend to have lower residence times than 75 s. However, in this study, the experimental residence time was set at a larger value to determine measurable changes in gas concentration.

Second, some air quality sensor systems depend on the ambient diffusion of gasses [10,11,14,16,17]. Hence, future experimental studies could explore reaction kinetics using a diffusion-based experimental setup measuring concentration changes over long periods of time in a closed chamber containing FDM-printed thermoplastics.

Third, the experiments were conducted at a constant ambient laboratory temperature (24.5 °C average). The FDM-printed baffles were at the same ambient laboratory temperature. Nevertheless, outdoor air quality measurement applications can regularly exceed 26.7 °C, especially when the sensor housing is directly exposed to sunlight. VOC off-gassing increases with increasing temperature [20,21,22,23,26,27], and VOC off-gassing is expected to increase with temperature as observed in the Antoine or Clausius–Clapeyron equation, desorption rate models, and diffusion coefficient in Fick’s law [19,21]. NO_x_ reaction rates with thermoplastics increase with increasing temperature [37,38]. Consequently, temperature’s effect on thermoplastic off-gassing and reaction rates should be evaluated in a further study.

Fourth, dry gas was used in the experiments, and relative humidity effects were not evaluated. However, chemisorption of water molecules is a known surface phenomenon [21,65] and may impact the off-gassing or reaction rates of thermoplastics used in air quality applications.

Fifth, while the aim was to explore the effects of recently FDM-printed thermoplastics on gas concentrations, the baffles were not reprinted between each set of gas experiments. As a result, the off-gassing rates and reaction rates presented in this work may underestimate the actual off-gassing rates and reaction rates of more recently FDM-printed thermoplastic components.

Finally, this study did not consider mass transfer effects for either the off-gassing or reaction analyses. Pariiskii et al. [37] discuss how polymer reactions with NO_x_ are non-uniform as the gasses diffuse through the solid. However, as reactions become more diffusion-controlled, mass transfer effects are expected to cause the reaction rates to decrease over time. Kwon et al. [21] showed that diffusion contributes to long-term off-gassing from polymers in vacuum systems. For both diffusive off-gassing and diffusion-controlled reactions, the thickness of the FDM-printed material would need to be considered. Battes et al. [23] showed a general relationship between an increase in thermoplastic thickness and an increase in VOC off-gassing rate. Future studies should evaluate the diffusive effects of FDM-printed components in air quality applications, including the effects of thermoplastic thickness and infill percent. This study considered a more limiting case where the reaction rates were assumed to be kinetically controlled and diffusion was assumed to be negligible, which should result in larger estimates for off-gassing and reaction rates. Despite these limitations, this paper’s results are a first step in exploring the effect of FDM-printed thermoplastic on measurements of common air quality gasses.

## 4. Conclusions

This study suggests that FDM thermoplastics exhibit no to limited interaction with CO and CO_2_ and a small interaction with NO and NO_2_. In air quality applications where all five trace gasses (CO, CO_2_, NO, NO_2_, and VOCs) need to be measured in a single sensor node, PLA, PETG, or PC are reasonable thermoplastic choices for FDM-printed housings or housing components. The temperature dependence of both off-gassing and reaction rates and how this would impact the gas concentrations need to be evaluated. A key finding is that VOC off-gassing of the FDM thermoplastics (except PVDF) could be important for low-cost air-quality sensor applications, particularly those operating in outdoor environments, focusing on low levels of VOCs, and using passive gas transport.

## Figures and Tables

**Figure 1 sensors-24-02610-f001:**
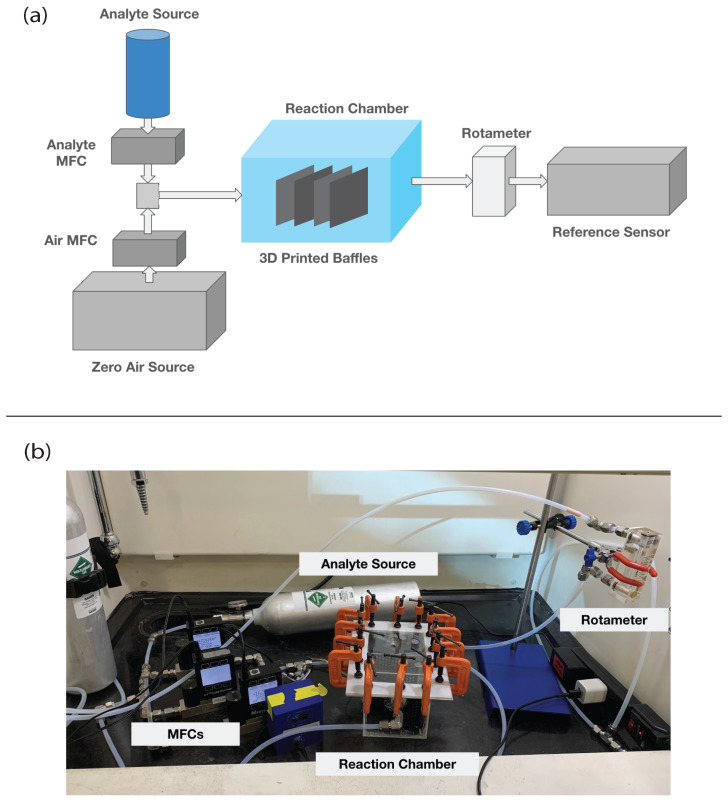
Experimental setup. Panel (**a**) shows a diagram of the experimental setup, and panel (**b**) shows an image of the experimental setup. Mass flow controllers (MFCs) regulate gas flow into a chamber that contains no baffles when measuring baseline concentrations and contains the baffles when measuring reaction or off-gassing. A rotameter measures chamber outlet flow that is directed to a research-grade gas instrument for gas concentration measurements.

**Figure 2 sensors-24-02610-f002:**
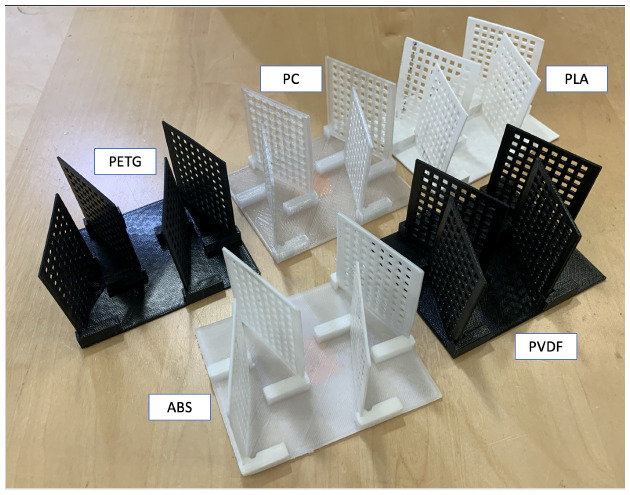
FDM-printed baffles for each of the five thermoplastics: polylactic acid (PLA), acrylonitrile butadiene styrene (ABS), polyethylene terephthalate glycol (PETG), polycarbonate (PC), and polyvinylidene fluoride (PVDF).

**Figure 3 sensors-24-02610-f003:**
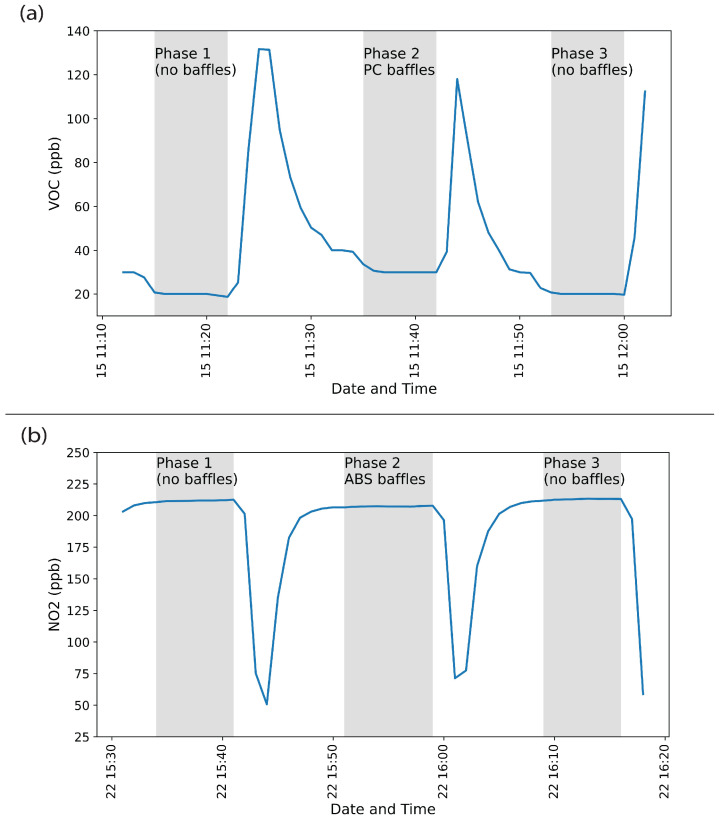
Concentration measurement time series graphs showing the three phases of the experiments. Concentration measurements from the research-grade instruments were averaged at one-minute intervals. Panel (**a**) shows a time series for a VOC off-gassing experiment with polycarbonate (PC), and panel (**b**) shows a time series of a NO_2_ reaction experiment with acrylonitrile butadiene styrene (ABS). Gray-shaded areas indicate the phases. The white areas show transitional periods where the lid of the chamber was removed to either add or remove the baffles, as well as the time allowed for the research-grade instrument to return to a steady state after changing between the phases.

**Figure 4 sensors-24-02610-f004:**
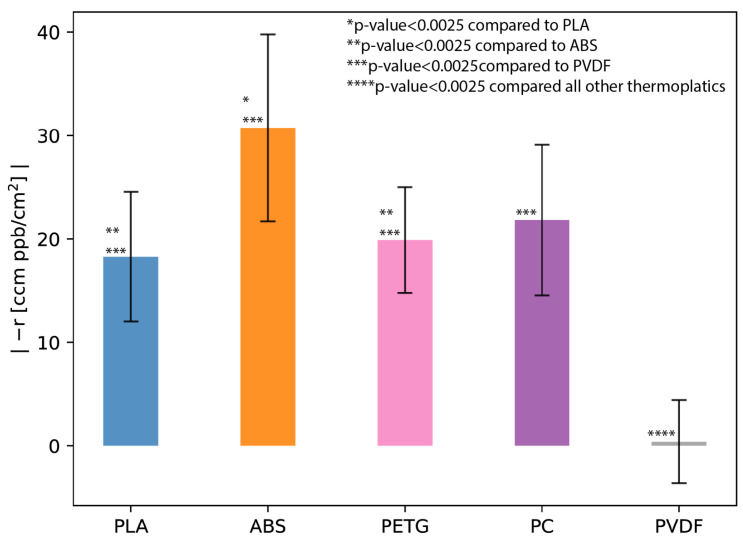
VOC off-gassing experiment results showing the absolute value of the reaction rate (reaction is negative for off-gassing) per surface area for the five thermoplastics—polylactic acid (PLA), acrylonitrile butadiene styrene (ABS), polyethylene terephthalate glycol (PETG), polycarbonate (PC), and polyvinylidene fluoride (PVDF). Error bars show the estimated error (Equation (Equation 2)) in the reaction rate calculated from measurement uncertainties. Asterisks show the significance testing results between the reaction rates for each thermoplastic.

**Figure 5 sensors-24-02610-f005:**
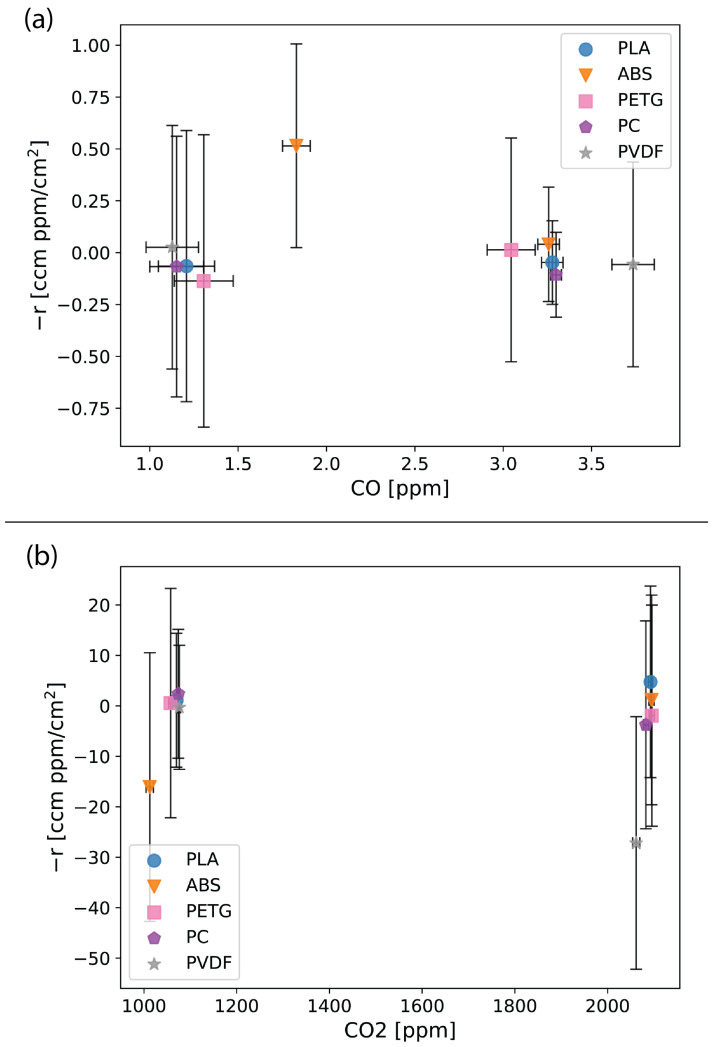
Reaction results for CO (**a**) and CO_2_ (**b**) with the five thermoplastics—polylactic acid (PLA), acrylonitrile butadiene styrene (ABS), polyethylene terephthalate glycol (PETG), polycarbonate (PC), and polyvinylidene fluoride (PVDF). Error bars show the estimated error (Equation (Equation 2)) in the reaction rate calculated from measurement uncertainties.

**Figure 6 sensors-24-02610-f006:**
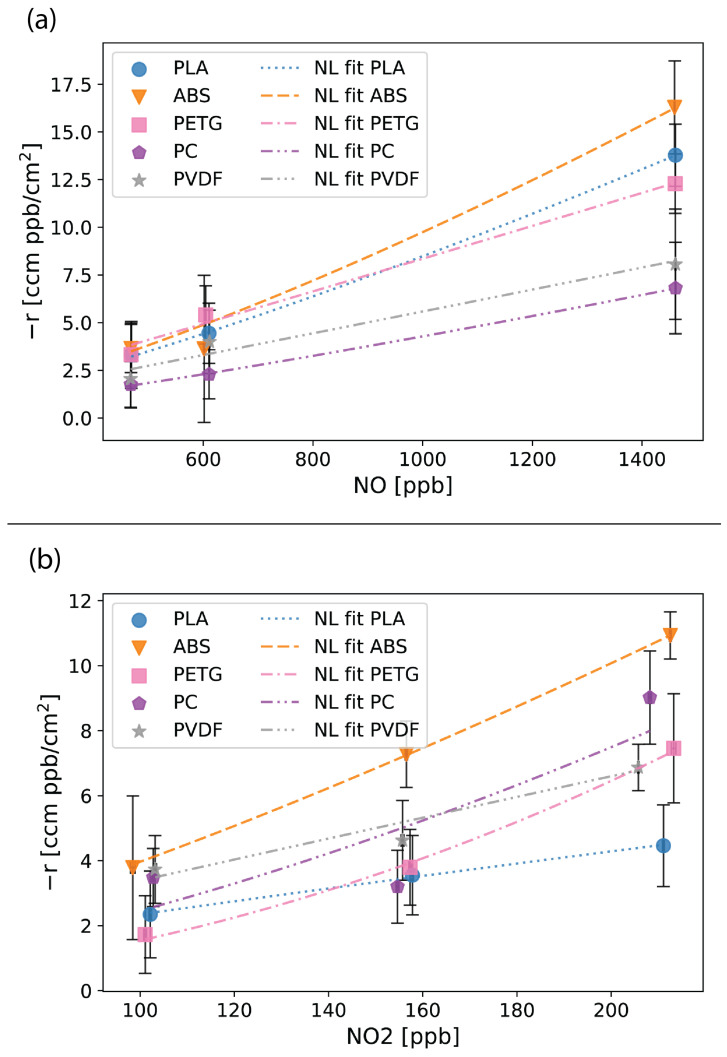
Reaction results for NO (**a**) and NO_2_ (**b**) with the five thermoplastics—polylactic acid (PLA), acrylonitrile butadiene styrene (ABS), polyethylene terephthalate glycol (PETG), polycarbonate (PC), and polyvinylidene fluoride (PVDF). Error bars show the estimated error (Equation (Equation 2)) in the reaction rate calculated from measurement uncertainties. The graphs include a nonlinear (NL) fit for the reaction kinetic equation (Equation (Equation 4)) for each thermoplastic.

**Figure 7 sensors-24-02610-f007:**
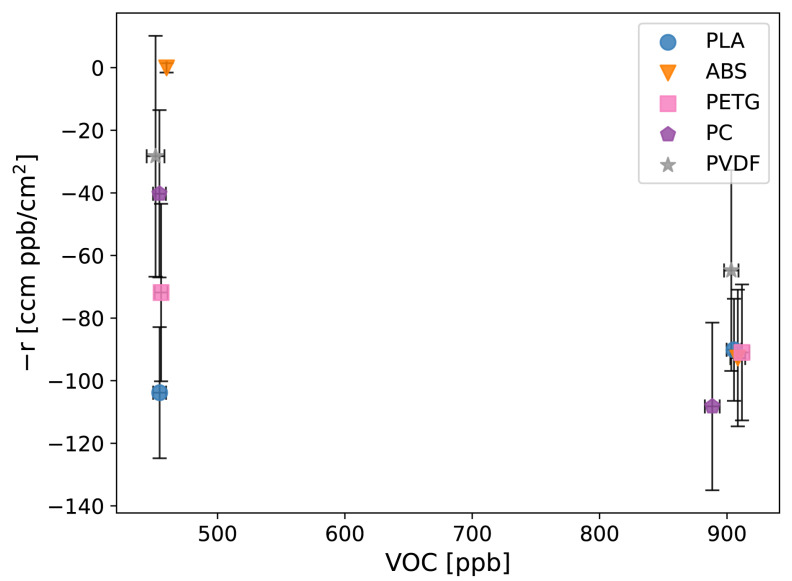
Reaction results for VOCs with the five thermoplastics—polylactic acid (PLA), acrylonitrile butadiene styrene (ABS), polyethylene terephthalate glycol (PETG), polycarbonate (PC), and polyvinylidene fluoride (PVDF). Error bars show the estimated error (Equation (Equation 2)) in the reaction rate calculated from measurement uncertainties.

**Table 1 sensors-24-02610-t001:** Studies that evaluated VOC emissions during FDM printing.

Study	Thermoplastic Filaments ^1^	VOC Detection Method (Off-Line)	VOC Detection Method (Real-Time)	Thermoplastic VOC Emission Ranking (Highest-Lowest)
Wojnowski et al. [28,29]	PLA, ABS, ASA, Nylon, and PETG		Proton transfer reaction time-of-flight mass spectrometry	ABS > PLA > ASA > Nylon > PETG
Floyd et al. [30]	ABS, PLA, PVA, HIPS, PCABS, Nylon, Bronze-PLA, and PET	Tri-sorbent sampling tubes	Photoionization detector (PID)	Bronze-PLA > PCABS > ABS > PVA > PET > Nylon > PLA > HIPS
Stefaniak et al. [31]	ABS and PLA	Silonite-coated canisters	PID	ABS > PLA
Kim et al. [32]	ABS and PLA	2,4-DNPH sorbent cartridges and absorbent charcoal tubes	PID	ABS > PLA
Davis et al. [33]	ABS, PLA, Nylon, HIPS, and PVA	Tenax TA mesh sorption tubes and 2,4-DNPH sorbent cartridges		Nylon > HIPS > ABS > PLA > PVA
Wilkins et al. [35]	PLA, ABS, and PETG		Semiconductor VOC sensor	Each thermoplastic filament emitted VOC during print—no ranking available ^2^
Weber et al. [34]	Nylon, PLA, and ABS	Absorption columns		Nylon > ABS and PLA ^3^

^1^ Thermoplastic acronyms: polylactic acid (PLA), acrylonitrile butadiene styrene (ABS), acrylonitrile styrene acrylate (ASA), polyethylene terephthalate glycol (PETG), polyvinyl alcohol (PVA), high-impact polystyrene (HIPS), polycarbonate-ABS thermoplastic alloy (PCABS), polyethylene terephthalate (PET). ^2^ The filament emitting the highest average VOC changed for each type of test (unenclosed, enclosed, and enclosed with the fan). ^3^ VOCs were emitted from both ABS and PLA, but the study did not report the magnitude of VOC emissions from either ABS or PLA.

**Table 2 sensors-24-02610-t002:** Research-grade instruments used to measure the concentrations of the atmospheric trace gasses and select specifications for each instrument.

Trace Gas	Instrument	Specifications ^1^
CO/CO_2_	TSI Q-Track with probe 982	Operating principle: electrochemical CO, NDIR ^2^ CO_2_
		Range: 0–500 ppm CO, 0–5000 ppm CO_2_
		Required flow rate: 500 ccm
		Resolution: 0.1 ppm CO, 1 ppm CO_2_
		Response time: <60 s CO, 20 s CO_2_
		Reference: [44,45]
VOC	TSI Q-Track with probe 984	Operating principle: PID ^3^
		Range: 10–20,000 ppb
		Required flow rate: 500 ccm
		Resolution: 10 ppb
		Response time: <3 s
		Reference(s): [44,45,46,47,48]
NO/NO_2_	Thermo Fisher NO_x_ Analyzer	Operating principle: chemiluminescence
		Range: 0–2000 ppb NO, 0–200 ppb NO_2_ ^4^
	Model: 42i	Limit of Detection: 0.40 ppb
		Required flow rate: 700 ccm (900 ccm recommended)
		Resolution/Precision: ±0.4 ppb
		Log interval: 60 s
		Reference: [49,50]

^1^ The CO electrochemical sensor has cross-sensitivities below 10% for other reducing/oxidizing chemical species [51]. The CO_2_ NDIR sensor is highly selective based on its specific spectral band [52]. The PID sensor demonstrates varying selectivity based on the VOC species being measured [46]. Instead, the TSI Q-Trak with probe 984 (PID sensor) reports an isobutylene-equivalent total VOC measurement for VOCs in the monitoring environment with ionization energy less than 10.6 eV (krypton lamp) [46,53]. Finally, the Thermo Fisher NO_x_ Analyzer exhibits high selectivity through chemiluminescence monitoring based on the characteristic reaction of NO with ozone [49,54]. ^2^ NDIR—Non-dispersive infrared. ^3^ PID—Photoionization detector. ^4^ NO_2_ range of 200 ppb due to molybdenum converter.

**Table 3 sensors-24-02610-t003:** FDM baffle and printer characteristics.

Thermoplastic Filament ^1^	Date Printed	Baffle Surface Area (±Error)	Extrusion Temperature Range	FDM 3D Printer
		**[${\mathrm{cm}}^{2}$]**	**[°C]**	
ABS	5/9/2023	538 (±3)	210–240	FlashForge Creator Pro 2
PLA	5/10/2023	538 (±3)	180–220	FlashForge Creator Pro 2
PETG	5/11/2023	538 (±3)	230–260	FlashForge Creator Pro 2
PC	5/12/2023	538 (±6)	250–270	LulzBot TAZ6
PVDF	6/18/2023	538 (±6)	245–265	LulzBot TAZ6

^1^ Thermoplastic acronyms: polylactic acid (PLA), acrylonitrile butadiene styrene (ABS), polyethylene terephthalate glycol (PETG), polycarbonate (PC), and polyvinylidene fluoride (PVDF).

**Table 4 sensors-24-02610-t004:** Summary of experimental conditions.

Gas (unit)	Flow Rate (±Error)	Off-Gassing Concentration	Reaction Target Concentrations ^1^	Temperature ^2^
	**[ccm]**	**[ppb|ppm]**	**[ppb|ppm]**	**[°C]**
CO (ppm)	1100 (±1.12)	0 (zero-air)	2, 4	24.2 (±0.3)
CO_2_ (ppm)	1100 (±1.12)	0 (zero-air)	1000, 2000	24.6 (±0.3)
NO (ppb)	1100 (±1.12)	0 (zero-air)	500, 1000, 1500	25.2 (±0.4)
NO_2_ (ppb)	1100 (±1.12)	0 (zero-air)	100, 150, 200	24.6 (±0.5)
Isobutylene (VOC) (ppb)	1100 (±1.12)	0 (zero-air)	400, 800	24.3 (±0.4)

^1^ Reaction experiment target concentrations were chosen based on the measurement ranges of the research-grade instruments listed in Table 2. While most applications for low-cost air quality sensors target lower ambient gas concentrations, Davydov et al. [36] utilized higher concentrations in order to observe polymeric degradation effects in a “reasonable time” and discussed that estimations of effects from lower gas concentrations could be extrapolated from the results at higher gas concentrations. ^2^ Each experiment was performed at the ambient temperature of the laboratory.

## Data Availability

The data presented in this study may be obtained from the authors upon reasonable request.

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
