# Peer review of "Effect of Three-Dimensional-Printed Thermoplastics Used in Sensor Housings on Common Atmospheric Trace Gasses"

_sensors, 2024, doi:10.3390/s24082610_

Round 1
Reviewer 1 Report
Comments and Suggestions for Authors
Line 40 “To the authors’ knowledge, no studies have examined carbon (CO)”: I think one word is missing, it should be "carbon oxide (CO)" and ideally it should be carbon monoxide and nitrogen monoxide instead of nitric oxide for NO.
Line 58 “PC and PVDF were chosen for this study because these are commercially available FDM thermoplastic filaments and are expected to have small reactivity with NO2 based on their chemical structures”: I would rather use the verb “added” than “chosen” to create a link with the previous sentence. Moreover, I understand the choice of NO2 but why did you not choose also O3 that is known to have a higher reactivity than NO2?
Line 83 “1500 cm3 acrylic chamber”: was the acrylic chamber sealed by some chemical or glue? I'm asking before having looked at the result of any zero air measurement. I may have my answer below.
Line 113-114 “This approach ensured that the baseline concentration measurements in phases one and three considered any interactions with the acrylic chamber.”: I've got my answer to my previous question, thank you.
Table 4 “Each experiment was performed at room temperature.”: that seems pretty high for room temperature, at least from my European point of view where we consider room temperature being around 20/22°C.
Line 140-141 “For ANOVA analysis, a p-value less than0.05 was considered statistically significant for ANOVA.”: I think only saying "for ANOVA" only one time should be enough.
Line 208-219 “Table 1 summarizes the results from previous studies that have examined VOC emissions during the FDM-printing process. This study differs from these previous studies in its focus on VOC emissions after completion of the FDM print. However, the relative magnitude of VOC emissions from the previous studies can still provide a source of comparison. Similar to the previous studies, this study also found that ABS had the highest VOC emission rate. However, this study found that PLA and PETG had similar VOC emission rates after FDM-print completion, while the Wojnowski study found that PETG had a lower VOC emission rate than PLA during the FDM-printing process. This difference could be due to the different conditions where this study considered VOC emissions after FDM-print completion while Wojnowski examined VOC emissions during the FDM-printing process. PVDF had the lowest VOC emissions, but no previous studies were found evaluating PVDF for comparison.”: I don't find this comparison between the during printing VS off-gassing emissions very usefull, in particular that it does not bring any information on the off-gassing process and that the experiment where not carried out with the same conditions. Somehow, it seems that the explanation of the differences (end of the paragraph) paraphrases the introduction to the paragraph. I do not get the point.
Line 236 “Figure 5 panels (c) and (d) indicate”: There is no panels (c) or (d) in Figure 5. You probably meant panels (a) and (b).
Line 275-276 “VOC off-gassing could be a particular concern if the sensor relies on passive gas transport and has a long residence time.”: Not only for passive sampler, also sensors which give real time measurement may suffer from interferences, drift or bad zero calibration. To a certain extent, also more sophisticated device such as GS-MS may suffer from this type of off-gassing as they accumulate pollutant during the pre-concentration phase.
Line 298-292 “Third, the experiments in this study maintained a constant, indoor ambient temperature (24.5°C average). Nevertheless, outdoor air quality measurement applications can regularly exceed 26.7°C, especially when the sensor housing is directly exposed to sunlight. Temperature can significantly influence both polymer off-gassing rates and reaction rates, making it important to assess this impact.”: what about temperature below 25°C?
Line 315-317 “In summary, the impact of the FDM thermoplastics on trace air-quality measurements should be considered as part of low-cost, air-quality sensor design.”: I found this overall conclusion sentence a bit too simplified.
Comments on the Quality of English LanguageNo particular comment, as already indicated in the recommendations for Authors.
Reviewer 2 Report
Comments and Suggestions for Authors
The authors studied how FDM-printed thermoplastics affect the measurement of trace atmospheric gases. The idea is innovative, and most people don't pay much attention to it. It has systematically tested several gases that are of greater concern to thermoplastic materials and environmental monitoring, which provides a certain reference value for researchers in related fields. I have some questions:
1. Why does PVDF release the least VOC compounds? Is it related to its high glass transition temperature?
2. What criteria were according to select the concentrations of nitrogen oxides and carbon oxides used in the experiment?
3. The glass transition temperature of PLA is lower than that of PC and PVDF, its toughness is stronger than PETG, and it does not interact with NO2. It is a candidate material for the sensor housings. Are there any other shortcomings of PLA as a candidate material as the housing material? If it exists, how to solve it?
4. Figure 2 provides a picture of the baffles. How to prepare baffles through 3D printing, and the specifications of the baffles should be briefly supplemented in the experimental section.
5. This work used baffles of the same size and number to evaluate the adsorption or reaction of five materials with the target gases. In actual application, how different are the thickness and surface area of ​​the sensor housing from that used in the measurement. In actual application, what is the estimated impact of the housing material on the air quality sensing?
Comments on the Quality of English LanguageThe English is good.
Reviewer 3 Report
Comments and Suggestions for Authors
Question 1: My understanding of your manuscript is that under the premise of several key assumptions, the effect of five common thermoplastics on the concentration of five common trace gases (CO, CO2, NO, NO2, VOCs) in the atmosphere is measured by the reaction rate of gases, so as to provide advice on the selection of housing materials for low-cost air quality sensors. In doing so, how did you make sure that several key assumptions raised did not affect the results of the real test? What are the main technical difficulties in this research that you proposed, how did you solve these difficulties, and have you considered introducing them?
Question 2: I noticed that you mentioned the surface area and extrusion temperature of the baffle in the introduction of the sample, but did not introduce the state of the baffle when entering the chamber. If your experimental design is to put the baffle after printing, have you considered the influence of parameters such as the temperature of the baffle? If it is placed after reaching a certain state, have you considered adding this parameter introduction? At the same time, will there be any potential factors that will affect the results of the baffle in this experiment when it reaches the put in state?
Question 3: I am very interested in the research-grade instruments you introduced. The minimum detection limit of the sensor required in the experimental design you introduced should be relatively low. In the detection environment with low concentration, will other interfering factors, such as airflow disturbance or the error of the sensor itself, cause interference to the detection of the sensor? Has any consideration been given to the introduction of a method for the detection of the steady state of gas concentration in the experimental design, and the reaction of thermoplastics with atmospheric trace elements over a long period of time?
Question 4: I noticed that the residence time of the constant airflow setting used in your experimental design is 75s. How was this time determined? It is suggested to add the correlation curve of gas concentration detection in the experiment to help observe the trend of gas concentration stabilization time more directly. I noticed that the input concentrations of the five gases were different in the reaction experiment. How was the reaction concentration of this gas determined?
Reviewer 4 Report
Comments and Suggestions for Authors
This study investigates how five different FDM-printed thermoplastics (ABS, PETG, PLA, PC, PVDF) affect the concentration of five common atmospheric trace gases (CO, CO2, NO, NO2, VOCs). The laboratory results show that the thermoplastics, except PVDF, exhibit VOC off-gassing. The results also indicate no-to-limited interaction between all of the thermoplastics and CO and CO2 and a small interaction between all of the thermoplastics and NO and NO2. 1. the ability of the reference sensor should be described, as its sensitivity, selectivity and resolution. 2. if the environmental temperature and humidity will influence the results? 3. the characters of materials (ABS, PETG, PLA, PC, PVDF) should be introduces, such as molecular structure, and grades. 4. it is better to use GCMS to calibrate the results.
Round 2
Reviewer 3 Report
Comments and Suggestions for Authors
Your answer is very perfect, and it solves the problem I raised very well.
Reviewer 4 Report
Comments and Suggestions for Authors
No further questions, since the authors didn't use GCMS to calibrate the results, I think this point is import to ensure the reliable of their results.